# pathways-ensemble-analysis v1.0.0: an open-source library for systematic and robust analysis of pathways ensembles

Lara Welder[1], Neil Grant[1], Matthew J. Gidden[1,2]

[1]Climate Analytics, Berlin, Germany
[2]International Institute for Applied System Analysis, Laxenburg, Austria

*Correspondence to*: Lara Welder (lara.welder@climateanalytics.org)

## Abstract

Ensembles of mitigation pathways, produced by multiple different models, are becoming increasingly influential as the world seeks to define climate goals and implement policy to meet them. In this context a range of open-source code has been developed to standardise and facilitate the systematic and robust analysis of mitigation pathways. We introduce a new open-source package, *pathways-ensemble-analysis* which provides an object-oriented framework for the key steps in analysis, describing its structure and providing an illustrative example of its use. By following the suggested application steps of the tool, a user can conveniently perform a systematic and robust analysis of pathway ensembles. This tool is therefore a further step which can help the community in conducting best practice in pathways-ensemble analysis.

## 1 Introduction

Energy and emissions pathways, such as those produced by Integrated Assessment Models (IAMs), are becoming increasingly influential as the world attempts to address the issue of global warming and reduce emissions rapidly towards net zero in line with the Paris Agreement (Weyant 2017, Krey 2014, Keppo *et al* 2021).

There are, however, many different future pathways which could comply with the Paris Agreement. Such pathways may vary across demographic, socio-economic and technological dimensions, meaning that there is a large solution space of possible low-carbon futures which merit consideration. There is therefore a need to understand how to compare and contrast different pathways (Grant *et al* 2020), as well as how to draw robust insights from a large number of pathways (Guivarch *et al* 2022b). This requires the analysis not of single pathways, but of a *pathways-ensemble* – a collection of multiple energy and emissions pathways.

The analysis of pathway ensembles has grown rapidly in recent years, largely due to the rise of scenario databases (Huppmann *et al* 2018, Byers *et al* 2022). These are databases containing a large number of pathways, often produced by a wide range of underlying IAMs. Such ensembles

were created to accompany the IPCC's Special Report on 1.5C, and again for the IPCC's Sixth Assessment Report. They have become influential sources of information on what the world needs to do to limit warming to 1.5C, and have been used by a wide range of actors. The rise of such databases has initiated a discussion about how to derive robust insights from them (Guivarch *et al* 2022b, Ferrari *et al* 2022).


The development of pathways ensemble analysis has been supported by standardised open-data and open-source code. In context of the submission and analysis process of scenarios for IPCC-related activities but also for model intercomparison projects, a standardised way of managing data structures with the open-source Python package *nomenclature* has been developed

(Huppmann *et al* 2024) which can help standardise the scenario data provided, enabling easier comparison of different pathways by using data templates (IAMC 2024). To analyse, validate and visualise the scenario data given in this data template, the open-source Python library *pyam* has also been developed (Huppmann *et al* 2021, 2023). The library includes a number of plotting options which enable a side-by-side comparison of models and/or scenarios with only small

amounts of additional coding required.

The tools developed so far provide standardised data reporting and analytical tools, which can help when analysing large number of pathways concurrently. However, there remains space to further develop tools for pathways ensemble analysis. In particular, the ability to filter pathways

to select a subset of a broader ensemble, the ability to identify illustrative pathways via a systematic approach, and the ability to visualise and plot key indicators of the ensemble as a whole, remain important tasks for which further tools can be developed.

Here we present a new Python-based open-source package, the *pathways-ensemble-analysis* or

*p-e-a*, tool (Welder and Grant 2023). This package provides these functions, improving the ability of the community to conduct systematic and robust analysis of pathways ensembles in a convenient way.

## 1.1 The use of ensemble analysis in the literature

Different forms of pathways ensembles analysis can be found in the literature.

### 1.1.1 Model inter-comparison exercises

Model-intercomparison projects are designed to investigate a specific research question with different models that have harmonised scenario parameters assumptions. In these, the pathways analysis can be performed 'in-situ', allowing for adaptations and iterations of model-scenario combinations. Insights can be obtained from within-ensemble agreement but should be caveated

if 'structural differences are not systematic and models share approaches or components' (Wilson *et al* 2021, Parker 2013).

Recent model-intercomparisons which have produced and analysed pathway-ensembles have explored the cost and attainability of meeting climate goals without overshoot (Riahi *et al* 2021),

the potential for good practice policies to close the emissions gap (van Soest *et al* 2021), the temperature implications of current mitigative efforts (van de Ven *et al* 2023), and to help determine the structural differences between models (Dekker *et al* 2023a).

## 1.1.2 Assessing a pathways-ensemble ex-situ

As well as 'in-situ' pathways ensemble analysis, it is also possible to conduct 'ex-situ' analysis.
'Ex-situ' refers to analysis of ensembles which have already been created, either for a specific research project or by combining together pathways from multiple different research projects. The ensemble is now being analysed after its creation to answer a given research question. The most obvious example is the ex-situ analysis of scenario databases collated and assessed by the IPCC.

Two examples of how to derive 'ex-situ' insights from a pathways-ensemble are statistically derived, stand-alone indicators and the analysis of illustrative pathways. Such an ensemble can be 'unstructured', in the sense of it not originating from a single model inter-comparison exercise but rather be a collection of different, individual projects that can 'give an indication of the spread of results in the literature' (van Diemen *et al* 2022).

- 'Stand-alone indicators' highlight an individual aspect of a pathways-ensemble based on statistical averages. For example, the median level of greenhouse gas reductions from 2019-2030 in a pathways-ensemble can be calculated as a 'stand-alone' indicator. Such indicators are valuable, but represent a statistical property of the ensemble, rather than a
single, self-consistent pathway that has a particular underlying scenario narrative. Examples of stand-alone indicators include key benchmarks on global emissions reductions provided by the IPCC (IPCC 2023), as well as the expansion rate of global renewable capacities to meet a climate goal (Climate Analytics 2023b) or emission reduction levels needed to keep a country on track with the Paris Agreement (Climate
Action Tracker 2023, Climate Analytics 2023a). We note that stand-alone indicators can also be used for in-situ analysis, as seen in Dekker et al (2023a), and also that scenario ensembles should not generally be seen as statistical ensembles, and thus the interpretation of medians or other quantiles of the distribution requires care (see Section 1.1.3 and the Conclusions for further discussion of this topic).

- 'Illustrative pathways' on the other hand, are single pathways extracted from the ensemble because they demonstrate particular dynamics which are of interest. They can be used to investigate the "implication of choices on socio-economic development and climate policies, and the associated transformation of the main [greenhouse gas]-emitting sectors"
that result from a particular set of assumptions / particular scenario narrative (Riahi *et al* 2022). Illustrative pathways have been used to communicate results in a wide range of settings (Riahi *et al* 2022, Smith *et al* 2023, Climate Analytics 2022).

To determine stand-alone indicators based on statistics or to select illustrative pathways from a
pathways-ensemble, analyses often start by applying a filtering process which returns a subset of pathways of particular interest for the analysis.

A simple example of a filtering process is the application of filters to ensure that the pathways display correct historical behaviour, also known as a 'vetting' process, (Guivarch *et al* 2022a). This filtered ensemble is then used further to determine 'stand-alone indicators', as for example emission reduction levels and levels of carbon dioxide removal, as well as five 'Illustrative Mitigation Pathways' (Riahi *et al* 2022).

The filtering process can also be applied more rigorously, for example by being informed by a political framework, as the Paris-Agreement, or feasibility, sustainability or ethical concerns, as for example about the technical potential for carbon storage (Grant *et al* 2022), the availability of sustainable biomass (Fuss et al 2018) or distributive justice concerning negative emissions (Minx et al 2018). Applying such filters can have a strong impact on the results, which highlights the importance of applying filters in a rigorous and systematic way (Achakulwisut *et al* 2023).

## 1.1.3 Challenges, risks and good-practices

In-situ model-intercomparison projects strive towards clean comparisons of pathway data for the specific research question they investigate. While they enable a focused exploration of a specific research question, they are however labour and computationally resource intensive and require access to input data, models and required hardware.

Performing "ex-situ" analysis on larger pathway ensembles pulls together a larger set of evidence. The potential benefits of using large ensembles include that they may better capture uncertainties, increase the salience, credibility and legitimacy of the information produced and is a way of building a comprehensive or representative picture of the knowledge produced by modellers (Guivarch *et al* 2022b). Such ensembles are nevertheless not a meaningful, random statistical sample that fully covers a potential solution space. Bias exists, for example through model fingerprints and / or an overrepresentation of multiple similar scenarios coming from the same model-intercomparison projects (Guivarch *et al* 2022b, Peters *et al* 2023). This can introduce confounding effects beyond the mechanism that an ex-situ analysis attempts to study.

Given the challenges and risks, Guivarch et al. propose a three-step approach for preparing and using ensembles of mitigation scenarios, which include
1. Pre-processing the ensemble, including quality control and vetting as well as reporting and potentially correcting bias,
2. Either
   a. transparently selecting scenarios from the ensemble, for example based on specific (un)desirable outcomes, plausibility criteria, or seeking to represent the diversity of the ensemble, or
   b. exploring the full ensemble, and
3. Providing users with efficient access to the information, including decision-support and communication tools and transparent and reproducible meta-analysis.

In addition to this, we highlight that when communicating statistical properties calculated from a pathways ensemble, it is important to highlight that these describe and parameterise the existing

'ensemble of opportunity' of (generally) normative scenarios, rather than a full statistical ensemble. As such, interpreting these values as indicative of probabilities, expected values or statistical ranges should be avoided.

## 1.2 Aim of the p-e-a package

Both the 'in-situ' and the 'ex-situ' pathways ensemble analysis share a number of common steps.
These are:
- the evaluation of criteria based on model results,
- an optional filtering process to select only a subset of pathways, and
- a well-laid out, if desired 'rated', side-by-side comparison of the remaining pathways with their evaluated criteria which can then be used for further analysis.

These steps should be guided by the above mentioned good-practices (Guivarch *et al* 2022b)

This paper introduces a Python-based workflow, *pathways-ensemble-analysis*, which standardises and automates these steps, building on existing work in the research community such as the Python library *pyam*. The workflow can thus support the analysis of model-
intercomparison projects and pathways ensembles by providing additional, easily obtained insights which provide a fast and, when guided by good-practices, well-laid out and comprehensible overview of the pathway ensemble of interest. This can be used in both *in-situ, ex-situ* and blended project setups, in which both elements are present.

The method of this workflow will be outlined in the next section and an application will be presented in the section after.

# 2 Method

In the Method section, we first illustrate the workflow of the Python package. Second, we provide a description of how the package is implemented.

## 2.1 Workflow


This section describes the developed workflow which derives a well-laid out, comprehensible overview of a pathways-ensemble. The workflow is implemented in an object-oriented manner in the open-source Python library *pathways-ensemble-analysis*.

Figure *1* visualises an illustrative workflow:

1. The analysis starts with extracting pathways data. Typically, these are obtained either from local files in a IAMC data format or are downloaded from a pathway database, as for example from the ones hosted by IIASA (Huppmann *et al* 2018) which can be conveniently
accessed using *pyam*. Typically, external data pre-processing routines are run on such datasets to address missing or faulty data. An example of missing but patchable data is if

the total use of bioenergy in the power sector is given and the use of bioenergy with carbon capture and storage (CCS) but the use of bioenergy without CCS is not provided. An example of faulty data is when the total electricity generation does not add up to the sum of its components which can be remedied by either recalculating the total or dropping redundant components. Once the data is pre-processed, it is passed on as a `pyam.IamDataFrame` object.

2. The next step is the definition and evaluation of criteria for each pathway. Examples of criteria are the emission reductions in 2030 with respect to a base year, the share of non-biomass renewables in 2050 in the power sector, the mean carbon sequestration via land use / biomass / fossil fuels over a given number of years, the maximum exceedance probability of a temperature limit or the magnitude of regional differentiation in a pathway. In this step, *pyam*'s filtering functions and a mixture of algebraic operations with *pyam* and *pandas* is being used to evaluate the criteria before finally returning a *pandas.DataFrame*.

3. The next step is to filter the pathway ensemble to select a subset of the initial ensemble. A filtering process drops pathways with criteria outside a given range from the ensemble. Examples of filters are to avoid overreliance on negative emissions from land use or bioenergy with CCS across a given time period. This optional step is of specific interest for ex-situ analysis of pre-existing pathways ensembles, and might be of lesser importance for model inter-comparison projects which can partly enforce these filters *a priori* in their scenario input parameters.

4. Having produced a filtered subset of pathways for analysis, the pathways can be rated along a range of criteria defined in step 2. The criteria used to rate pathways can be those which were used to filter the database, and/or additional used-defined criteria. The usage of the rating function is twofold. On the one hand, the function can be used to normalise the criteria, for example map them to values from 0 to 1, and in this way improve the readability of the final output plots. On the other hand, the function can be used to rate the criteria of each pathway based on normative preferences. Simple examples of rating functions are:
    a. To have a high share of non-biomass renewable electricity generation: $x \rightarrow x$
    b. To have a low share of fossil electricity generation: $x \rightarrow 1 - x$

5. The such rated criteria are then available for visualisation. Outputs can for example be visualised with a heatmap which displays the rated criteria with the filtered pathways sorted based on their overall rating.

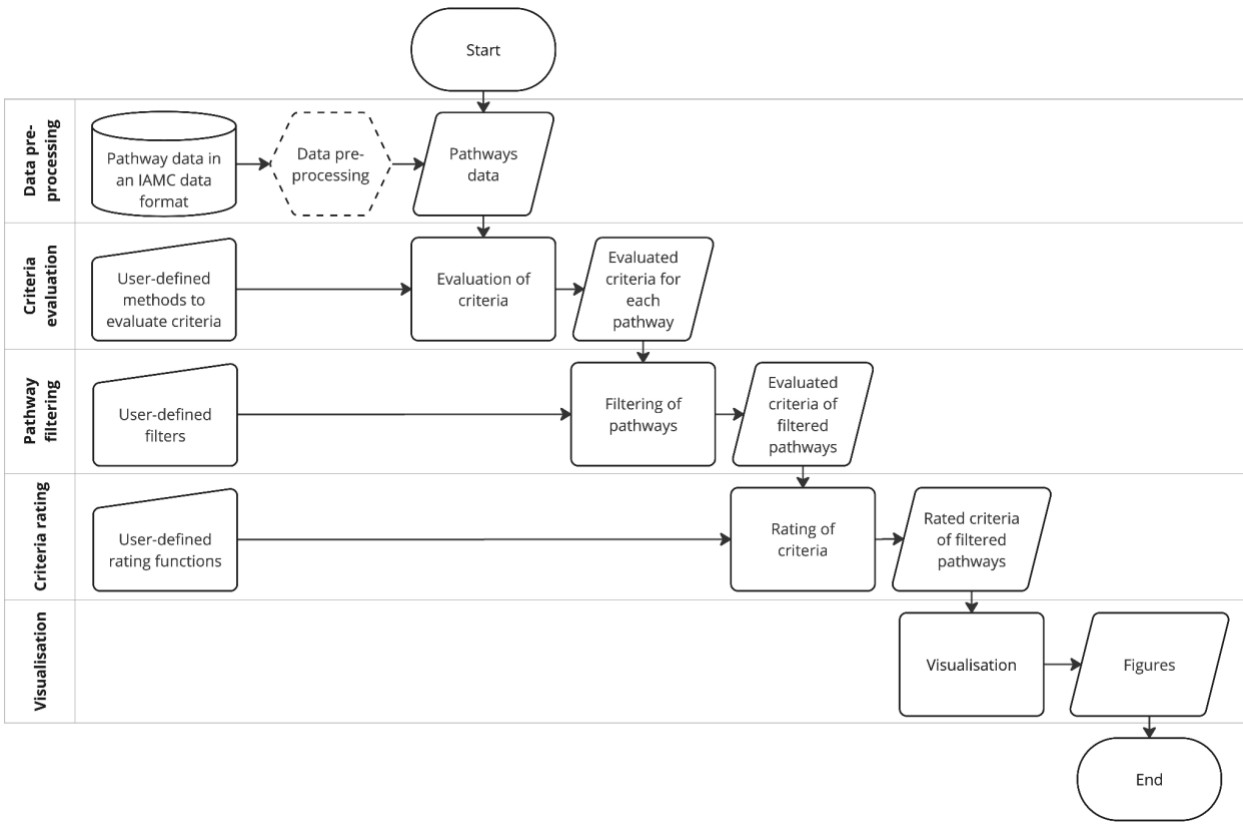


Figure 1: Flowchart setup provided for the pathways-ensemble analysis. The "Data pre-processing" process in dashed lines is in theory optional but is advised to be addressed with external programming routines.

## 2.2 Package description

The Python library containing the object-oriented setup of the workflow is structured as followed:

- In the `evaluation` module, the core methods `get_values`, `filter_values`, `rate` and `filter_rating` are located, which process the pathways data, user-defined criteria and other user-defined input data as visualised in the Figure *1*.
- In the `criteria` module, classes for criteria are implemented which, at a minimum, contain a `criterion_name`, `rating_function`, `rating_weight`, `region` and a

  `region_aggregation_weight` as class parameters and a `get_values` and a `rate` method as class functions. The `criteria` module contains two sub-modules:
  - In the `base` module, the following criteria classes are currently implemented:
    - `Criterion`, the basic criterion class which other criteria inherit from.
    - `SingleVariableCriterion`, which evaluates the value of a variable for a

      given year and region.
    - `AggregateCriterion`, which evaluates the aggregate of a variable, for example the average / min / max, for given years and a given region.
    - `ChangeOverTimeCriterion`, which evaluates the change of a variable for a given year and region with respect to a reference year.

    - `ShareCriterion`, which evaluates the share of a component on the total for a given year and a given region.

- `CompareRegionCriterion`, which takes a pre-defined criterion (for example the share of renewables in the electricity mix) and two regions, and calculates a metric which compares the value of the criterion in each region. Currently the comparison can be either a `subtract` or a `divide` operation.
    - In the `library` module, criteria for specific, reappearing use cases are implemented (pre-set parameters can be changed by the user), examples are:
        - `Mean_CarbonSequestration_Fossil`, evaluates the average amount of global, fossil CCS across the years 2040 to 2060. The rating function is informed by literature values on the potential of CCS (Guivarch *et al* 2022a, Budinis *et al* 2018).
        - `Mean_CarbonSequestration_Biomass`, evaluates the average amount of globally sequestered carbon via bioenergy with CCS across the years 2040, 2050 and 2060. The rating function is informed by estimates of the global potential of sustainable negative emissions from bioenergy with CCS (Fuss *et al* 2018).
        - `Mean_CarbonSequestration_LandUse`, evaluates the average, global carbon dioxide emissions from afforestation and reforestation across the years 2040, 2050 and 2060. The rating function is informed by estimates of the global potential of sustainable / feasible potential of negative emissions coming from afforestation and reforestation (Fuss *et al* 2018, Grant *et al* 2021).
        - `Mean_Biomass_PrimaryEnergy`, evaluates the average, global amount of biomass-use in primary energy across the years 2040, 2050 and 2060. The rating function is informed by literature values on the sustainable, technical potential of bioenergy (Creutzig *et al* 2015, Frank *et al* 2021).
- The `plot` module is intended for providing plotting methods to the user. Currently three main plotting methods are provided here. The first, called `heatmap`, enables the visualisation of the pathways-ensemble for the criteria of interest. The second, called `compare_ensemble`, allows multiple different pathway ensembles to be compared using box plots. The third one is inspired by recent work (Dekker *et al* 2023a) and displays criteria values in form of a `polar_chart`.
- The `utils` module contains a number of utility methods used in other modules.
- A `tests` module is provided to ensure the quality of the code and support the continuous integration and development of new code.

# 3 Application

In this article, we demonstrate with one example how the *pathways-ensemble-analysis* repository can be used in the analysis of pathway ensembles. In this example, we use the package to identify a filtered subset of pathways from the IPCC AR6 scenario database (Byers *et al* 2022), highlight the impact of filtering on ensemble statistics, e.g. on stand-alone indicators, and identify an illustrative pathway for further investigation. Additional examples are briefly described in the last subsection, as for example a recreation of the IPCC AR6 vetting process (Guivarch *et al* 2022a) and a model fingerprint analysis, in the style of recently published work (Dekker *et al* 2023a). The

code to reproduce all of the presented analysis can be found in *notebooks* folder in the git repository of the package.

## 3.1 Input data to the workflow

The raw data which serves as input to this ensemble is the AR6 scenario database (Byers *et al* 2022). This provides 97 1.5°C compatible pathways which are the starting point for our analysis. This selection is in itself already a filtering step, but one that can easily be achieved with the *pyam* library.

We conduct an analysis using eight user-defined criteria. We distinguish between primary criteria and secondary criteria. Primary criteria are used to filter the database, directly excluding pathways which have particular behaviour in order to select a subset of pathways for analysis. Secondary criteria are not used directly in the filtering process, but are still used for rating and visualising the ensemble and support the selection an illustrative pathway of interest. Generally, it is up to the user to decide which criteria to use for a filtering step, and which to use in a rating step. The criteria are described in Table 1.

Table 1: Primary and secondary criteria used in the example

| | CRITERIA | FILTER THRESHOLD | SOURCE | MODULE / CLASS |
|---|---|---|---|---|
| **PRIMARY** | A / R deployment (2040-2060 average) | < 3.6 GtCO$_2$ / y | (Grant *et al* 2021) | `library` |
| | A / R deployment (2050-2100 average) | < 4.4 GtCO$_2$ / y | (Grant *et al* 2021) | `library` |
| | BECCS deployment (2040-2060 average) | < 5 GtCO$_2$ / y | (Fuss *et al* 2018) | `library` |
| | Regional differentiation on GHG mitigation (in 2030) | Mitigation (developed regions) > Mitigation (developing regions) | Author judgement | `ChangeOverTime Criterion` + `CompareRegionC riterion` |
| **SECONDARY** | Reduction in fossil fuel production/use by 2030 (relative to 2020) | - | - | `ChangeOverTime Criterion` |
| | Share of renewables in the power sector (in 2030) | - | - | `ShareCriterion` |
| | Fossil CCS deployment (2040-2060 average) | - | - | `library` |
| | Primary biomass demand (2040-2060 average) | - | (Creutzig *et al* 2015) | `library` |

The filters of the primary criteria have been used (alongside others) to identify a Paris-compatible set of pathways in a recent analysis (Climate Analytics 2023b).

The secondary criteria are not used for filtering but are used to obtain further insights into the pathways-ensemble. In this example, the aim is to focus on pathways which rapidly reduce fossil
fuel demand, based on deployment of renewables and limited reliance on biomass or fossil CCS. Such a focus could be justified by the precautionary principle (which would suggest faster emissions cuts), or with reference to the potential sustainability/feasibility concerns relating to biomass (Creutzig *et al* 2015) and CCS (Grant *et al* 2022).

## 3.2 Filtering of the ensemble and its impact on stand-alone indicators

Applying this filtering process to the IPCC's AR6 scenario database (Byers *et al* 2022) reduces the number of 1.5°C compatible pathways from 97 to 30 pathways.

Figure 2 shows the impact that the filtering has on the secondary criteria, using the `compare_ensemble` plotting function.

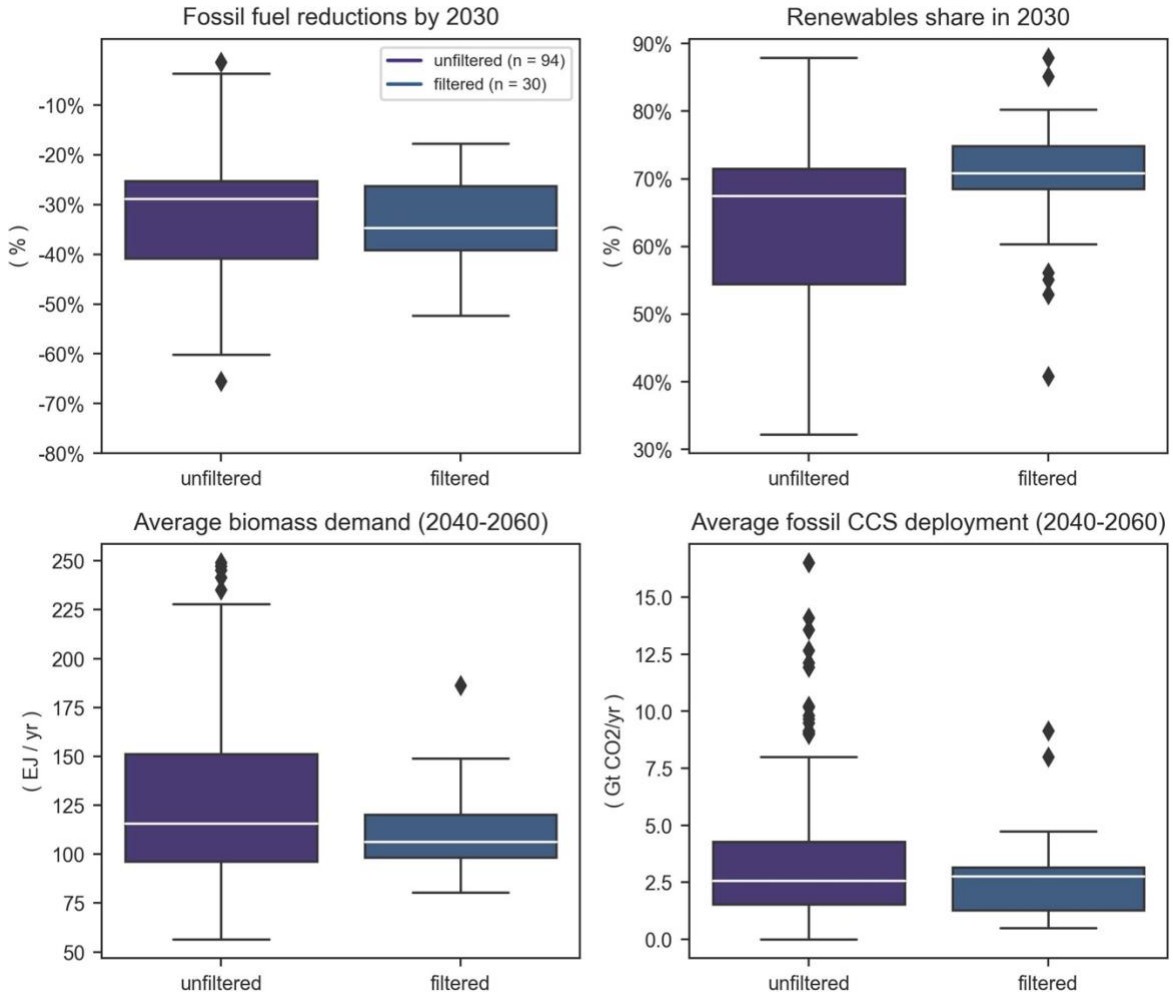

Figure 2: The impact of filtering on a selection of variables of interest

In this example, filtering the pathways ensemble to reduce reliance on future CDR leads to greater reductions in fossil fuel production/use by 2030 (a 35% reduction from 2020 levels, rather than a 29% reduction seen in the unfiltered ensemble). This greater action is driven in part by accelerated renewables deployment – with renewables making up 71% of the global electricity mix in 2030, up from 67% in the unfiltered ensemble. Reduced reliance on future CDR also corresponds to reduced reliance on biomass as an energy carrier.

The changes in the median of these stand-alone indicators are sometimes minor, but there are nevertheless large changes in the overall ensemble range. This is particularly evident in the renewables share, biomass demand and fossil CCS deployment indicators, where the filtering process excludes those pathways with the lowest renewables deployment, and highest reliance on biomass / fossil CCS. As interquartile or total ensemble ranges are often provided alongside

the median as influential key statistics (Rogelj *et al* 2018, Riahi *et al* 2022), this highlights the potential influence of filtering on the results of pathway analysis. Given the key focus at the moment on the role of fossil fuels in mitigation pathways (Achakulwisut *et al* 2023), the influence of the filtering on a key benchmark such as fossil fuel reductions also shows the critical importance of considering filtering as part of a pathways ensemble analysis.


The such filtered pathways-ensemble can now be used to determine 'stand-alone' indicators, such as the median and ranges visualised in Figure 2. If more insights into the ensemble are desired, a side-by-side visualisation, coupled with an optional rating step, can be performed.

## 3.3 Rating and visualising the pathways-ensemble

A side-by-side comparison with normalised criteria values, for example ranging from 0 to 1, can support the analysis of how the different pathways achieve a 1.5°C compatible transformation pathway. The p-e-a's `rate` and `heatmap` plotting function can be used to facilitate this. If it is of additional interest to identify illustrative pathways for further analysis, these can be selected based on the ratings of each criterion.


We rank four main criteria to illustrate the differences between pathways. These criteria, first introduced as secondary criteria in Table 1, are shown in Table 2 with their rating functions. Rating functions have two main dimensions. First is whether the function is selecting for low or high values of the criterion. In this example, we select for low levels of biomass, fossil CCS and total

emissions (negative rating functions), with high levels of renewables (positive rating function). The second is the sensitivity of the rating function to the criterion values. By weighting the value of x more highly (e.g. `lambda x: np.clip(2*x - 1, 0,1)`) and applying threshold values, the rating function can increase the selectivity of the analysis to this variable. In the above example, values under 0.5 would score zero, and then every increase of 0.01 above this would increase

the score by 0.02. In this way, very tailored filters can be developed that select and highlight particular behaviour.

The developing of rating functions is an inherently normative process, but one which gives a high degree of control over which criteria to rate, and the relative importance of each criterion. If this

is transparently communicated, this flexibility and control is a key strength of the p-e-a.

Table 2: Rating criteria for the analysis

| | CRITERIA | RATING FUNCTION | RATIONALE |
|---|---|---|---|
| **RATED CRITERIA** | Reduction in fossil fuel production/use by 2030 (relative to 2020) | `-x` | We want to select pathways with the deepest reductions (so the lowest value of x). |
| | Share of renewables in the power sector (in 2030) | `np.clip(`<br>`2*x - 1,`<br>`0, 1`<br>`)` | Selecting pathways with the highest renewables share. Clipping the function to range from 0 to 1 over the 50-100% renewables share increases the selective power of this criteria. |
| | Fossil CCS deployment (2040-2060 average) | `np.clip(`<br>`1 - ((x-3.8)/(8.8-`<br>`3.8)),`<br>`0, 1`<br>`)` | Pathways with under 3.8 $GtCO_2$/yr of fossil CCS score 1. Pathways with >8.8 $GtCO_2$/yr of fossil CCS score 0. Thresholds are taken from the IPCC's feasibility assessment (Guivarch *et al* 2022a). |
| | Primary biomass demand (2040-2060 average) | `np.clip(`<br>`1 - ((x-50)/(150-`<br>`50)),`<br>`0, 1`<br>`)` | Pathways with under 50 EJ /yr of biomass demand (~current levels) score 1. Pathways with >150 EJ/yr of biomass demand score 0. Thresholds taken from IPCC's feasibility assessment (Guivarch *et al* 2022a). |

Having rated the pathways across the criterion of interest, we can visualise the pathways using the `heatmap` function. This function produces a heatmap, in which each column represents an individual pathway, and each row represents a user-defined criterion of interest. The function then calculates the aggregated rating for each pathway across the criteria, and assigns the pathway a total rating. Highest rated pathways are plotted at the left, with the pathway rating declining from left to right. The `heatmap` function gives the option to also plot criteria which are of interest, but are not used in the overall rating itself.

Figure 3 shows such a heatmap for the filtered set identified using the criteria in Table 1 (the 15 highest-scoring pathways out of the 30 pathways which pass the filters are shown). The pathways are rated and ordered according to the four secondary criteria of interest. Therefore, we are identifying pathways which both:

a) Pass the filters which are used as strict exclusion criteria
b) Have rapid reductions in fossil fuels in the near-term, driven primarily by renewables
deployment, with limited reliance on biomass and fossil CCS deployment

The heatmap also provides further insights into the model dynamics. For example, we can see that a few REMIND-MAgPIE pathways have a relatively low fossil CCS deployment and low average biomass demand, pointing at high wind and solar electricity shares in power generation without the need for fossil CCS. The shown COFFEE pathway has the highest share of renewable electricity generation which is however linked to a strong reliance on biomass demand. We can further observe that the displayed WITCH pathways have AFOLU emissions within the sustainability limits, while being more reliant on biomass, both in term of general demand as well as average BECCS deployment.



Figure 3: Heatmap that enables identification illustrative pathways

## 3.4 Selecting an illustrative pathway from the ensemble

As mentioned in the introduction section of this work, illustrative pathways can be extracted from the ensemble to demonstrate particular dynamics of interest. They can be used to investigate the "implication of choices on socio-economic development and climate policies, and the associated transformation of the main [greenhouse gas]-emitting sectors" that result from a particular set of assumptions / particular scenario narrative (Riahi *et al* 2022).

The process we applied so far has identified pathways which pass the defined exclusion criteria and promote rapid emissions reductions in the near-term, driven primarily by renewables deployment, with limited reliance on biomass and fossil CCS deployment. The first two pathways on the left side of the heatmap comply with these criteria particularly well – with having the highest rating across the ensemble – and are therefore candidates for further analysis. It is of interest to note that these are in fact two of the three 1.5ºC compatible illustrative mitigation pathways selected by the IPCC AR6 for further analysis (Riahi *et al* 2022).

The identification of illustrative pathways with differently chosen socio-economic developments and climate policies can be identified in a similar manner, using differently specified criteria.

## 3.5 Additional examples

The tool can be flexibly applied to investigate different characteristics of pathway ensembles. In the following, we briefly show two such examples. A detailed derivation and description of these examples can be found in the repository of the tool.

### 3.5.1 Vetting process

The IPCC AR6 vetting process (Guivarch *et al* 2022a) can be recreated in a straightforward manner with the tool. In this process, pathways that have historical energy and emission values outside of an acceptable range are being dropped from further analyses. Figure *4* displays the vetted historical criteria where the legend indicates how many pathways have information on the vetted criteria and how many of these remain in the ensemble after the filtering process.

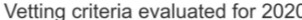

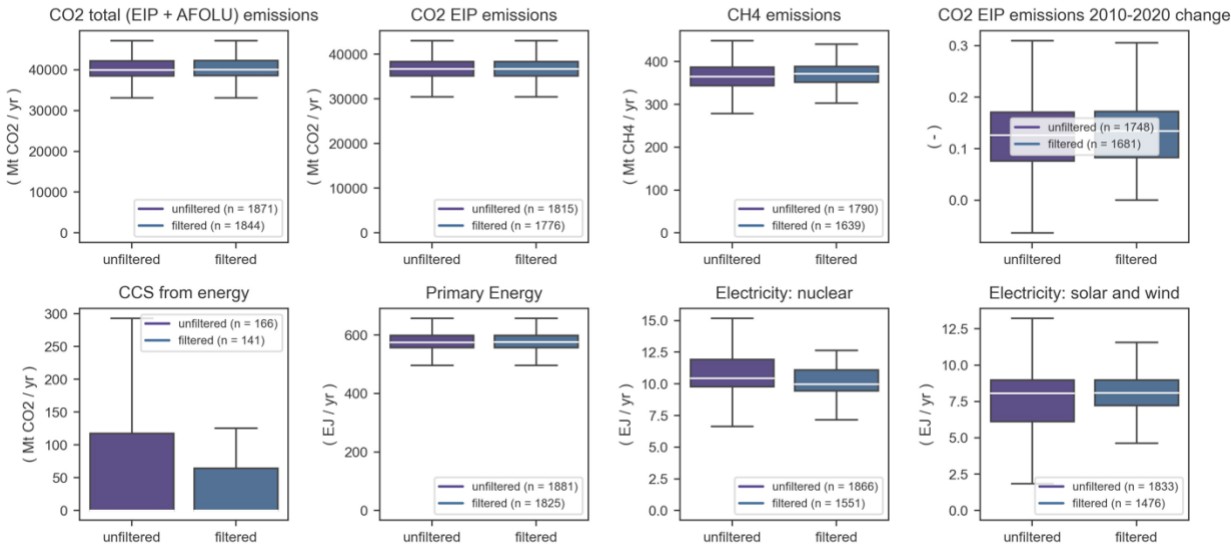


Figure 4 Filtering based on historical data vetting as done in the IPCC AR6 vetting process (Guivarch et al 2022a).

### 3.5.2 Fingerprint analysis

Inspired by recently published work on energy model fingerprints in mitigation scenarios (Dekker *et al* 2023a), the `polar_chart` plotting function can display statistical characteristics of the

chosen criteria / indicators, see Figure *5*.

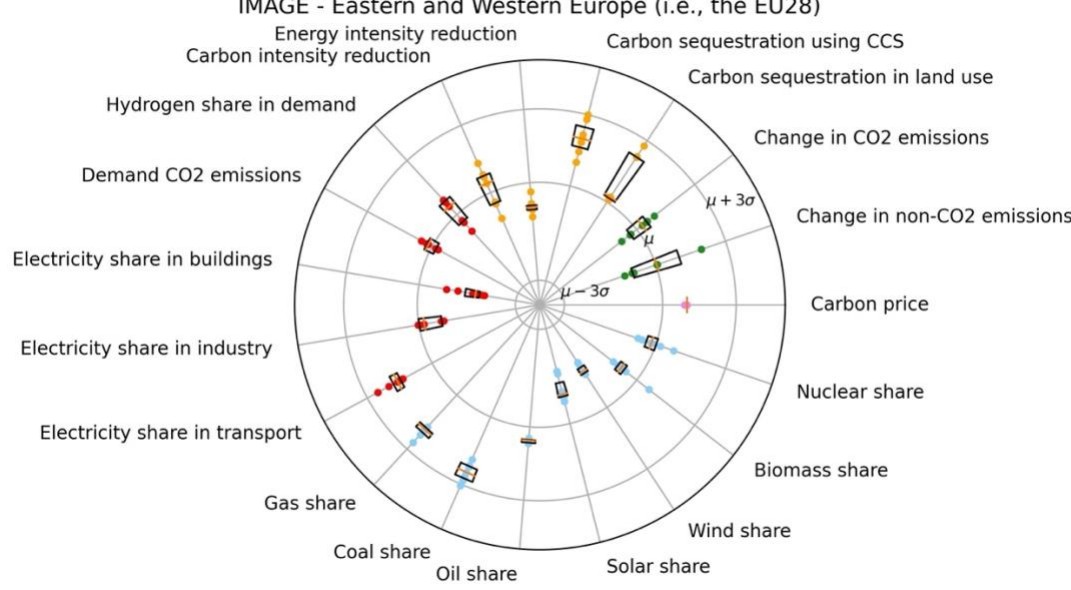

Figure 5 Example of a polar chart plot created with the pathways-ensemble analysis tool (inspired by recently published work on energy model fingerprints in mitigation scenarios (Dekker et al 2023a)).

# 4 Conclusion

The open-source library presented in this work provides the research community a tool to perform analyses of pathways ensembles. The library utilises and expands existing work of the community, specifically the *pyam* library, guaranteeing compatibility with current data standards and coding practices as well as an easy use.

The open-source availability on Gitlab provides transparency to the implemented method and is aimed to encourage the community to contribute and further expand the library. A testing module is integrated to support the continuous integration and development of new code.

       The object-oriented implementation of the core code of the library provides the user of the code
with the ability to design the analysis in a flexible manner, for example by setting the parameters of predefined criteria freely or by having the option to easily define new criteria as needed. It furthermore shortens otherwise implemented code significantly, resulting in concise and easy to write code blocks, which provides a good overview over the analysis and therefore convenience to the user.

       The library has a wide range of applications, including pathways ensemble analysis in model inter-comparison exercises or deriving ex-situ insights from (unstructured) pathways ensembles, for example to determine stand-alone statistical indicators or illustrative pathways. For this purpose, the library provides key functionalities commonly used in ensemble analysis. These include the
definition of criteria of interest and the evaluation, filtering and rating of these criteria, as well as visualisation functions which can help demonstrate the impact of filtering and rating.

       The impact of the filtering and rating operations are relevant to be cognisant of at almost all steps of such analyses. One example to highlight is the calculation of stand-alone, statistical indicators,
such as the level of fossil fuel reduction that complies with the Paris Agreement. The simple application provided in this work, which reduces the reliance on future CDR and therefore implies greater levels of ambition in the near-term, is already an example of this.

       Limitations to both the dataset and the method for processing these datasets in such analyses.
The scenario data itself can have missing or faulty data, the solution space is not statistically representative and therefore the calculation, and interpretation, of statistical indicators challenging. While working with illustrative pathways is not affected by the latter, the selection process to getting to these pathways is always influenced by the user-defined criteria with their filtering and rating functions.

       Nevertheless, literature also points out the benefits of using large ensembles "ex-situ", for example that they may better capture uncertainties (Guivarch *et al* 2022b). Under the premise that the underlying scenario dataset, with its bias and the choice of criteria, filters and rating functions, is processed with good-practices, for example clearly communicated, the functionalities
of the *pathways-ensemble-analysis* tool provide a foundation for performing a transparent, robust and systematic analysis of a pathways-ensemble. This library could be used in future community

endeavours such as the construction and evaluation of new IPCC scenario databases, model intercomparison projects, and the ex-situ analysis of IPCC databases to provide key metrics such as CDR and emissions reduction requirements. Criteria with predefined rating functions and filters could be discussed and standardised across the community.

Future work can be identified when reviewing recent work on pathways ensemble analysis in literature (Smith 2022, Guivarch *et al* 2022b, Dekker *et al* 2023a, 2023b).

- While filtering is a key step in determining robust insights into a pathways-ensemble, the structure of the ensemble should also be reflected upon critically. Here, one example is whether the calculation of stand-alone indicators should be weighted by the frequency with which a particular model features in the pathways-ensemble. This could help avoid models with a specific fingerprint and a high (or low) occurrence from being over- (or under-) represented in the insights derived from the ensemble. The *pathways-ensemble-analysis* ensemble could be extended such that the calculation of stand-alone indicators accounts for their relative representation in the overall ensemble. At the same time, models with a high level of occurrence in the pathways-ensemble could still provide a statistically relevant distribution of pathways, in which case weighting by model frequency may be less appropriate.
- Literature also provides inspiration for new analysis and plotting routines, such as for identifying model fingerprints by analysing criteria for individual models or by determining cluster of pathways with distinct characteristics (i.e., criteria). The *pathways-ensemble-analysis* could be used further in this endeavour, with an illustrative example provided on the repository.

# Code and data availability

The general *pathways-ensemble-analysis* GitLab repository is available under the MIT licence at https://gitlab.com/climateanalytics/pathways-ensemble-analysis. Version v.1.0.0 of the *pathways-ensemble-analysis* repository, which is presented in this paper, is available under GitLab and archived on Zenodo (Welder and Grant 2023). Version v.1.1.0, which includes an updated version of the input data and scripts to run the model and produce the plots for all the simulations presented in this paper, is available on GitLab and Zenodo as well (Welder and Grant 2024).

# Author contributions

LW conceptualised the workflow, set up the initial software code basis and wrote the manuscript draft; NG contributed to the conceptualisation of the workflow, expanded the software code basis, lead on the shown application and wrote the manuscript draft; MG contributed to the conceptualisation of the workflow and reviewed and edited the manuscript.

# Competing interests

The authors declare that they have no conflict of interest.

# Financial support

This research has been supported by the IKEA Foundation.

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
