# Peer review of "pathways-ensemble-analysis v1.1.0: an open-source library for systematic and robust analysis of pathways ensembles"

_EGUsphere, 2024_

## Author Response (AR1)

**REPLY 1**

**Summary:**

The paper describes a new package that helps users to analyze projections of integrated assessment models (IAMs). It facilitates filtering of pathways, identifying illustrative ones based on user-defined criteria and also a few plotting routines. Such tools are much needed to standardize analysis in this field and also make them (and researchers) more efficient. Thus, I am happy that the authors invest the time to make a well-documented open source package for this purpose.

*Thank you very much for this positive feedback.*

The manuscript could be made better if it would be more explicit about the exact contribution the package is after (standardization, convenience, etc.) and linked to that what the novelty and target audience exactly are – as some of the functionalities of the package are probably of less use to the more experienced pathway database users.

*We have highlighted the contributions of the package in the abstract and conclusion of the manuscript. As experienced pathway database users, we find the convenience the package provides of particular use (much shorter, more concise code that does not need to be rewritten for each new analysis).*

Also, a bit more demonstration in terms of example code and visualizations would help, with better link to other related work and existing packages.

*We have added two additional examples with visualizations into the manuscript. In terms of example code, we have added a reference to the notebook folder in the git repository.*

**General comments**

1. In the introduction section (1) I think it would help to describe a bit more what are the advantages and disadvantages of in-situ versus ex-situ ensemble analysis. I realize that they are categorically different, but a little more detail on these two approaches can help contextualize this work and why you have chosen to focus on an ex-situ comparison and what the limitations are of this. For example, in-situ analyses is partially crippled by being highly time/labor consuming and require a lot of model specialists, and also are rarely fully clean comparisons. While on the other hand, they do provide the 'cleanest' comparisons between scenarios: piling scenarios with different assumptions or from different projects together may lead to ill distributions and "noise" beyond the mechanism that one aimed to study. Hence, a middle road, with in- and ex-situ being hand-in-hand should always be taken. This discussion and beyond can maybe contextualize the current work work a bit.

   *We have included additional remarks in the introduction in a new subsection.*

2. Added to that, I would say that the contrast between what is in-situ and ex-situ is not always as sharp as the authors seem to suggest. While indeed, the fingerprint work in Dekker et al. (2023) uses specifically designed diagnostic scenarios to indicate model differences, methodologically it makes use of 'stand-alone indicators' from a scenario ensemble (albeit produced in a single project) to detect model behavior, quite similar to what the authors describe in 1.1.2.

   *Yes, we agree that stand-alone indicators can be applied to both types of analysis. We differentiate between "in-situ" and "ex-situ" via the possibility to do adaptations and iterations of model-scenario combination. While stand-alone indicators are a useful tool for in-situ pathways ensemble analysis, they are one of the two core techniques to analyze ex-situ*

*which is why we have highlighted them under the ex-situ section. We have added a caveat highlighting that they can also be used in in-situ analysis.*

3. Related to point [1]. A notable disadvantage of using pathways ensembles in an ex-situ manner is often the lack of a good statistical interpretation of the distribution. Especially when applying too may filters, I suspect that non-representative statistics start to appear or even empty solution spaces. I realize this is a shortcoming in this field in general, but maybe the authors can contextualize their work by discussing this in the conclusions. Especially since the compare_ensemble function emphasizes medians and quantiles.

   *We have added additional discussion pieces on this matter to the manuscript.*

4. Could the authors specify a bit sharper what gap / problem the authors are exactly addressing? Is it mostly convenience/increasing efficiency of researchers, or also a form of standardization (i.e., avoiding everyone doing their own thing), etc.?

   *See our response under „Summary".*

5. Why is it important to first specify the evaluation criteria (step 2) and only then start filtering? I could imagine that a user would also want to explore the opposite: first filter and then explore what characterizes these pathways. Some indicators would not be very statistically different from the full database, some indicators would suddenly collapse onto a small range consequential to the filtering. Incorporating a functionality that allows insights in that latter mechanism could be a nice addition to this package.

   *To first filter and then explore the criteria is also possible. If the "first filter" is based on meta data or a raw data series, pyam can be applied for the filtering. If the indicator is more complex, it could be described as a criterion and then be filtered for. The remaining set of pathways can then be further explored. The application of the code is very customizable in this regard.*

6. It would be useful to demonstrate a bit more what the package can do in terms of visualizations. For example, three types of plots are mentioned, but the polar charts are not demonstrated.

   *We have added a brief example with a polar chart and another example which shows how the package can be applied in a straightforward manner to recreate the AR6 vetting process.*

7. I believe the filtering functionality in itself it not novel in the sense that many researchers in our field do that themselves already. However, what would be very useful, is if the package holds some literature-based thresholds for filters based on for example what are regarded as feasible rollout rates of nuclear energy, CCS and CDR. We see in the AR6 database quite a big range in such developments that may or may not be feasible. It would be a form of 'pre-cooked' or 'suggested' filters that would save researchers a lot of time, and also facilitate standardization of such filters between different papers.

   *There is a number of 'pre-cooked' criteria with suggested rating functions available in the library module of the package (see 2.2 Package description). The rating functions are informed by literature-based thresholds. We would be happy to expand this list of criteria.*

8. Maybe point (6) is also related to what the authors feel are the target audience for this package. Could the authors expand on this? Tech-savvy colleagues may do some of these things themselves, and also do the visualization themselves.

   *See our response under „Summary".*

9. Would it be useful to, similar to the criteria selection (see point (6)), have 'pre-cooked' rating functions in regard to the normalized criteria values in section 3.3?

*See 7.*

10. The section on illustrative pathways could be expanded. For example, could you show all Riahi 2022 IMPs expressed in the same heatmap as figure 3 and show what this package reveals beyond those already well studied pathways.

*Thank you very much for this suggestion. We originally considered to add such a piece of analysis but decided to go for a simpler example instead to focus on the demonstration of the code.*

11. Could you express more how this package links to the ecosystem of other relevant packages currently available? Maybe using example code when linking to pyam?

*We have added mentions of pyam into the manuscript.*

12. The discussion/conclusion section could be extended a bit:
    1. Shortcomings of this package (or the methods behind it)
    2. Shortcomings of pathway ensembles in general
    3. How these type of analyses (filters, etc.) could be standardized in the community

*We have added remarks to the conclusion section.*

**Small comments**

- I would suggest using the 'fingerprint' analogy instead of the 'footprint' analogy in various instances throughout the manuscript. Fingerprint are telling of a model's or pathway's attributes, while a footprint is a form of pressure exerted onto the climate system or the database, for example.

*We followed your suggestion.*

**REPLY 2**

The authors present a Python package for filtering and classifying scenario modelling results from IAMs and related types of models, and for analyzing and visualizing the resulting scenario ensembles. Like the pyam package it is based on, the package looks like a very useful tool that has the potential to streamline and reduce the time required to complete the types of tasks that it covers. By its nature, it does not present anything that is substantially novel in terms of scientific content, but the package looks like it can contribute to better science by reducing the risk of errors, and by freeing up time for more scientifically valuable tasks that in turn can contribute better and more accurate scientific analyses. This is of great value in a field like analysis of scenario model results and scenario ensembles, where writing code or performing manual operations to process and visualize data makes up a significant part of the overall workload.

*We thank the reviewer for this assessment.*

The authors explain the background and need for their package succinctly and well (with one significant exception, see below). The package itself and how to use are adequately explained and the authors provide a useful example of a typical intended workflow. It Is clearly laid out how to obtain the source code and how to use it. The paper is in my mind nearly ready for publication, with one significant addition:

The discussion in section 1 of the scientific background for the package does have one significant shortcoming that I think the authors should address in order to be published: The package aims to simplify the work of creating statistical ensembles or make subselections from existing ensembles, but without much of a discussion of the weaknesses and pitfalls of making statistical analyses of such ensembles. In particular, scenario ensembles are generally not statistical ensembles. The distribution of values of a variable across scenarios in an ensemble is not necessarily a good indication of the distribution of that variable in the real world or across possible futures. This remains the case even after performing the type of filtering that pathways-ensemble-analysis does unless great care is taken in designing filters that correct any biases in the selection of models and the results produced by the models. But nevertheless statistical measures such as the scenario average or scenario spread of a variable are frequently used to represent expected values, ranges or even probabilities that a given variable might fall outside of a certain range.

The authors should discuss this issue more explicitly, to make sure the reader knows what caveats must be taken when interpreting the results of analyses on scenario ensembles. At the moment this is only mentioned in passing in the main text, and then briefly mentioned at the very end (line 414). There is a useful discussion as well as references to relevant case examples in the paper cited as Guivarch 2022b, but the authors don't use the citation in that context. Another paper that discusses the issue to some extent is https://www.nature.com/articles/s44168-023-00050-9. But I urge the authors to also include a proper explanation and discussion of the issue, not just cite these papers in passing.

*Thank you for this comment. We have added a "challenges, risks and good-practices" subsection to the introduction and made this point more explicit in the conclusion.*

Apart from this, it would be useful if the authors could add a few more examples of the charts that the package can produce. The example workflow and heatmap plot currently in the paper are very useful, but it would be good to also see an example of each type of plot that the package can produce. There wouldn't need to be a full workflow example for each plot. A small plot gallery would suffice, just to let the readers see what the package can produce without having to install and test each plotting function. This is entirely optional, but I think would be a very useful addition.

*We have included two additional, brief examples. This includes the third plotting option (polar_chart) from the plotting module of the package.*

---

## Author Response (AR2)

Dear Dr. Müller,

Thank you very much for your reply.

We have updated the title and added the reference within the sentence.

Best wishes on behalf of the author team.